

# Phase space compression of a positive muon beam in two spatial dimensions

A. Antognini[1,2⋆], N. J. Ayres[1], I. Belosevic[1], V. Bondar[1], A. Eggenberger[1],
M. Hildebrandt[2], R. Iwai[1†], K. Kirch[1,2], A. Knecht[2], G. Lospalluto[1], J. Nuber[2],
A. Papa[2,3], M. Sakurai[1], I. Solovyev[1,4], D. Taqqu[2] and T. Yan[1,4]

**1** Institute for Particle Physics and Astrophysics, ETH Zurich, 8093 Zurich, Switzerland
**2** PSI Center for Neutron and Muon Sciences, 5232 Villigen PSI, Switzerland
**3** University of Pisa and INFN, 56126 Pisa, Italy
**4** School of Physics and Center for Excellence in High Energy Physics and Astrophysics,
Suranaree University of Technology, Thailand

⋆ aldo@phys.ethz.ch , † iwai@frib.msu.edu

## Abstract

We present the first demonstration of simultaneous phase space compression in two
spatial dimensions of a positive muon beam, the first stage of the novel high-brightness
muon beam under development by the muCool collaboration at the Paul Scherrer In-
stitute. The keV-energy, sub-mm size beam would enable a factor $10^5$ improvement in
brightness for precision $\mu$SR, and atomic and particle physics measurements with pos-
itive muons. This compression is achieved within a cryogenic helium gas target with a
strong density gradient, placed in a homogeneous magnetic field, under the influence
of a complex electric field. In the next phase, the muon beam will be extracted into
vacuum.

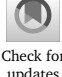

# 1 Introduction

Muons are fundamental particles belonging to the second generation of the Standard Model and can be copiously produced at high-intensity, medium energy accelerator facilities. They have a mass of $106\,\text{MeV}/c^2$, i.e. approximately 200 times the electron mass, and a lifetime of $2.2\,\mu\text{s}$. Muons are hence sufficiently long-lived for precision experiments and at the same time, adequately short-lived to precisely study their decays, making them a unique probe at the intensity and precision frontiers of particle physics [1], as well as for materials science [2].

Large numbers of (polarized) muons ($10^8\,\mu^+/\text{s}$) can be obtained from the decay of charged pions produced by a high-intensity proton beam of about 500 MeV to few GeV energies hitting a solid target, e.g. made of graphite [3,4]. The relatively long lifetime of muons allows their transport at energies of just a few MeV over distances of several tens of meters, from the pion production target to a dedicated experimental setup. This enables high-precision, low-background investigations. To achieve high muon rates, transport beam optics with a broad acceptance are employed, albeit at the expense of beam quality. This results in typical beam spots of centimeter-size, a divergence exceeding 100 mrad, and a MeV-energy spread [5].

For numerous experiments at the high-intensity frontier [6,7], it is advantageous to use a beam with an energy of a few MeV, spread over several centimeters, to suppress pile-up events. However, other experiments require muon beams with a small transverse phase space and low energy. Among the latter are investigations using muon-spin resonance ($\mu$SR) techniques [8] to study novel quantum materials (magnetic, superconducting, van der Waals, etc.), which are often available only in sizes of square millimeters or smaller. For instance, smaller sample sizes allow for the application of much larger uniaxial strain, opening new possibilities for the in-situ manipulation of quantum phases. The microbeam may become ultimately also a new tool for spintronic device developments that use the muon as a very sensitive local magnetic probe. For example, 3D-resolved studies of the induced spin-polarization in the active layers of various spintronic and/or super-spintronic devices could be performed [9]. This microbeam also paves the way for novel particle physics experiments that require precise control of the muon beam. For example, it could significantly enhance ongoing experiments aimed at searching for the muon electric dipole moment, where the beam must be coupled into a well-defined orbit suitable for storage within a solenoid [10]. Moreover, this new low-energy, high-brightness muon beam can be leveraged to generate innovative cold muonium beams with small transverse sizes and high intensity. This advancement could have a profound impact on precision experiments in gravity [11] and laser spectroscopy [12,13].

At the Paul Scherrer Institute (PSI), the muCool collaboration is developing a fast ($\lesssim 8\,\mu\text{s}$) cooling technique for continuous positive muon beams of few MeV energy using a cryogenic helium gas target featuring strong electric and magnetic fields following the seminal work of [14]. Conventional beam cooling techniques, such as stochastic cooling [15] or electron cooling [16] cannot be applied for muon beams due to the relatively short muon lifetime. An even faster cooling technique is being developed for positive muons at the Rutherford Appleton Laboratory (RAL) and J-PARC that utilizes muonium formation and re-ionization with a pulsed laser [17,18], but this is preferably applied to pulsed muon beams.

In this paper, we present the demonstration of the muon beam compression integral to the muCool scheme, which is designed to reduce the phase space of a typical positive muon beam by a factor of $10^9$ with an efficiency of up to $10^{-4}$, i.e. a brightness enhancement of up to $10^5$. This achievement marks a significant advancement toward the realization of the innovative muCool beam, which aims to produce a beam with tunable energy in the keV regime, with an energy spread of less than 100 eV and a sub-millimeter transverse size.

In Secs. 2 and 3 we introduce the complete muCool scheme, and the principle of the muCool cooling, respectively. In Sec. 4 we describe the experimental setup with emphasis on the

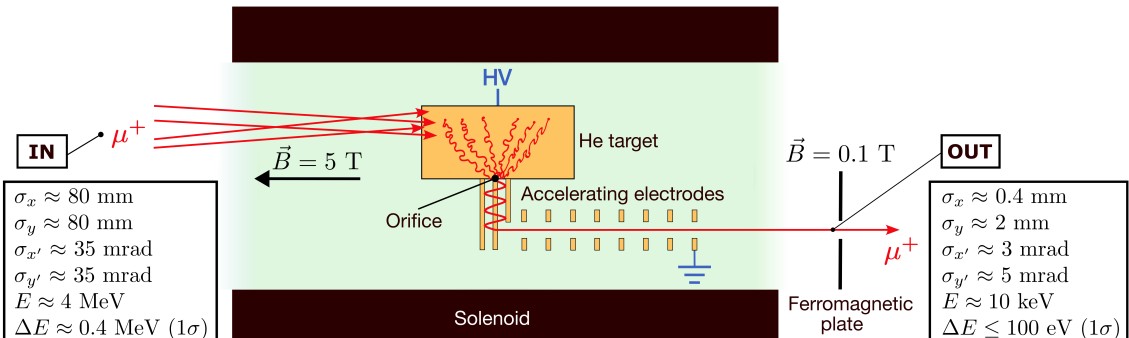

Figure 1: The muCool scheme: A positive muon beam of 4 MeV kinetic energy is directed into a solenoid and brought to a stop in a helium gas target at cryogenic temperatures, where compression (cooling) occurs. The compressed muons are subsequently extracted with eV-level energies from the target through an orifice and re-accelerated to keV energies along the magnetic field. Upon reaching a ferromagnetic plate equipped with a small aperture, the muons are extracted into a field-free region. To give a flavor of the compression achievable with the muCool scheme, we include the expected output beam parameters from preliminary simulations (before the magnetic shield) [19], to be compared with the input beam parameters based on the High Intensity Muon Beam (HIMB) [20, 21].

target that we used to demonstrate simultaneous compression in two directions. Moving on to Sec. 5, we present the measured time spectra and compare them to simulations to demonstrate muon beam compression.

## 2 The muCool scheme

A "standard" continuous muon beam, transported from the pion production target at an energy of 4 MeV (28 MeV/c momentum) with a few percent energy spread, is coupled into a cryogenic helium gas target at approximately 10 mbar pressure and placed in a 5 T solenoidal magnetic field, as illustrated in Fig. 1. This so-called surface muon beam has about 100% longitudinal polarization [22]. The target features a vertical temperature gradient from 6.5 K to 22 K, thus a corresponding He gas density gradient, and a complex electric field. Within the target, the muons undergo a rapid deceleration process, transitioning from MeV energy to just a few eV, rendering them highly sensitive to the electric field inside the target. The electric field and the He density distribution are precisely engineered to guide the dispersed muons, scattered across a large portion of the target volume, into a sub-millimeter spot within just a few microseconds. From this spot, the muons are extracted through a windowless orifice in the target and then re-accelerated electrostatically along the magnetic field lines toward a ferromagnetic plate, which serves to abruptly terminate the magnetic field. This magnetic shield features a small aperture that allows the muons to pass through, thus providing a muon beam in a field-free region. From there, the muons can be easily transported and focused to subsequent experimental setups or additional acceleration stages.

This paper focuses on experimentally validating muon beam compression within the helium gas target. For simplicity, we demonstrate this compression using a target without the orifice and with a muon beam of lower momentum (13.6 MeV/c instead of 28 MeV/c). This approach increases the stopping probability within the target, thereby improving the signal-to-noise ratio of our measurements without compromising the core findings of this study.

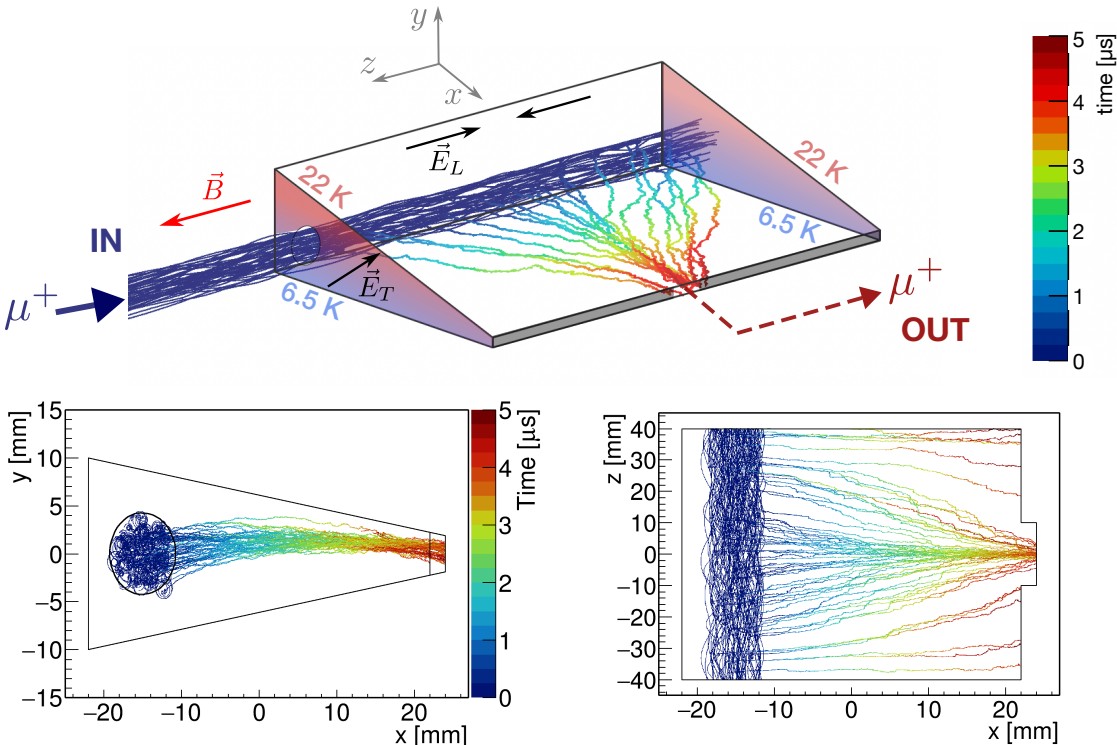

Figure 2: (Top): Sketch of the muCool target geometry with the magnetic field $\vec{B}$ and electric field $\vec{E}$, featuring a transverse component $\vec{E}_T$ in the $xy$-plane at 45° with respect to the $x$-axis and longitudinal components $\vec{E}_L$ in $\pm z$-direction pointing to the target mid-plane. The vertical temperature gradient is also indicated. The simulated muon trajectories demonstrate fast simultaneous compression in transverse ($y$) and longitudinal ($z$) directions. The color scale represents the time relative to the muon entering the target. (Bottom, Left) Simulation of the muon trajectories projected onto the $xy$-plane demonstrating compression in the $y$-direction while drifting in the $x$-direction toward the tip of the target. The dashed circle indicates the initial position of the muons at time $t \approx 0$, just after slowing down in the target. (Bottom, Right) Simulation of the muon trajectories projected onto the $xz$-plane demonstrating compression in the $z$-direction while the muons drift in the $x$-direction.

## 3 The principle of the muon beam compression in the helium gas target

A fraction of the muon beam entering the target is stopped in the helium gas at around 10 K temperature and a pressure of 10 mbar, resulting in a cylindrically shaped stop-volume of about 10 mm diameter and 50 mm length. In a few microseconds, these stopped muons move to a sub-millimeter spot at the target tip under the combined influence of the E-field, the vertical density gradient and the B-field as shown in Fig. 2 (Top). To understand the working principle of this compression, it is illustrative to consider the drift velocity $\vec{v}_D$ of a positive muon inside the helium gas in the presence of E and B-fields [23]:

$$\vec{v}_D = \frac{\mu|\vec{E}|}{1 + \frac{\omega^2}{\nu_c^2}} \left[ \hat{E} + \frac{\omega}{\nu_c} \left( \hat{E} \times \hat{B} \right) + \frac{\omega^2}{\nu_c^2} \left( \hat{E} \cdot \hat{B} \right) \hat{B} \right], \tag{1}$$

where $\mu = \frac{e}{m_\mu \nu_c}$ is the muon scalar mobility, $\nu_c$ is the average $\mu^+$-He collision frequency, $\omega = \frac{eB}{m_\mu}$ is the cyclotron frequency of the muons with mass $m_\mu$. $\hat{E}$ and $\hat{B}$ are the unit vectors along $\vec{E}$ and $\vec{B}$, respectively. Compression of the muon beam requires the drift velocity to be position-dependent i.e., $\vec{v}_D = \vec{v}_D(x, y, z)$, so that muons stopped at different locations drift in different directions. This position dependence is obtained by engineering a position-dependent E-field, depending only on the $z$-coordinate ($\vec{E}(z)$), and a position-dependent gas density $n(x, y)$, such that $\vec{v}_D(x, y, z) = \vec{v}_D[\vec{E}(z), n(x, y)]$.

The electric field, sketched in Fig. 2 (Top), has *transverse* ($xy$-plane) and *longitudinal* ($z$-direction) components $\vec{E} = \vec{E}_T + \vec{E}_L$, with *transverse* and *longitudinal* referring to the orientation relative to the B-field. The homogeneous *transverse* electric field $\vec{E}_T = \frac{E_T}{\sqrt{2}}(\hat{x} + \hat{y})$ leads to $\vec{E}_T \cdot \hat{B} = 0$, making the last term of Eq. (1) vanish. Under the influence of $\vec{E}_T$ the muons drift at an angle $\theta$ with respect to the $\hat{E}_T \times \hat{B}$-direction given by:

$$\tan \theta = \frac{\nu_c}{\omega}. \qquad (2)$$

Because of the vertical density (temperature) gradient (see Fig. 2 (Top)), both the collision frequency $\nu_c$ and the drift angle $\theta$ are smaller at the top of the target and larger at the bottom. Hence, muons stopped at the top of the target drift downward while moving to the target tip, and muons stopped at the bottom drift upward, respectively.

This behavior is confirmed by GEANT4 simulations [24–26] of the muon trajectories (see Fig. 2 (Bottom, Left)) that rely on simulated field maps and density distributions, as obtained from COMSOL Multiphysics simulations [27], and that account for energy-dependent elastic $\mu^+$ - He collisions. These collisions were implemented using energy-dependent transport cross sections obtained from energy scaling of proton data, as detailed in [26,28,29]. The GEANT4 simulations also account for other collisional processes such as the muonium formation and muonium ionization [26,30], but these processes are marginal in the 1 to 10 eV kinetic energy range at which the muon beam compression takes place.

Compression in $z$-direction, shown in Fig. 2 (Bottom, Right), is caused by the *longitudinal* electric field $\vec{E}_L$ pointing toward the target mid-plane. Indeed, following Eq. (1), a purely longitudinal E-field, $\vec{E} = \vec{E}_L \parallel \vec{B}$, results in a drift velocity along the B-field ($\vec{v}_D \sim \vec{B}$) as $\vec{E} \times \hat{B} = 0$ and $\vec{E} \cdot \hat{B} = E_L$. Similarly, for a purely longitudinal electric field anti-parallel to the B-field we find that $\vec{v}_D \sim -\vec{B}$.

Summarizing, the combination of transverse ($\vec{E}_T$) and longitudinal ($\vec{E}_L$) electric fields together with the vertical density gradient yields a simultaneous compression in the $y$- and $z$-directions as the muons drift in the $x$-direction. We refer to this simultaneous compression as *mixed transverse-longitudinal* compression or simply *mixed* compression.

Compression of a muon beam in either longitudinal or transverse direction was already demonstrated in earlier experiments: the longitudinal compression in a target at room temperature with a longitudinal electric field [31], and the transverse compression in a target at cryogenic temperature with density gradient and transverse E-field [28]. The goal of this study is to demonstrate mixed compression in a cryogenic target as shown in Fig. 2.

## 4 Experimental setup

The target that we realized to demonstrate the mixed compression is shown in Fig. 3 (Right). It is obtained by folding a 50 μm thick Kapton foil, shown in Fig. 3 (Left), around a 3D printed plastic frame and sealing it with a cryogenic and electrically isolating glue (Stycast epoxy [32]). To define the electric field in the target, the Kapton foil is lined with 18 μm thick gold-coated

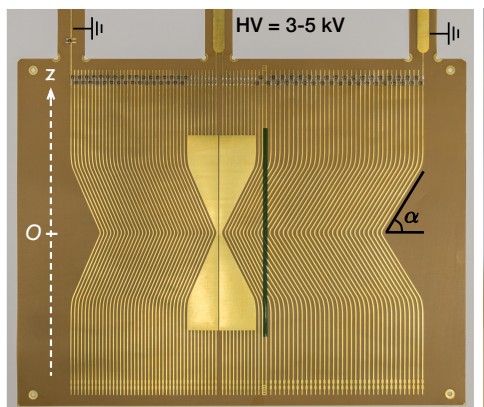
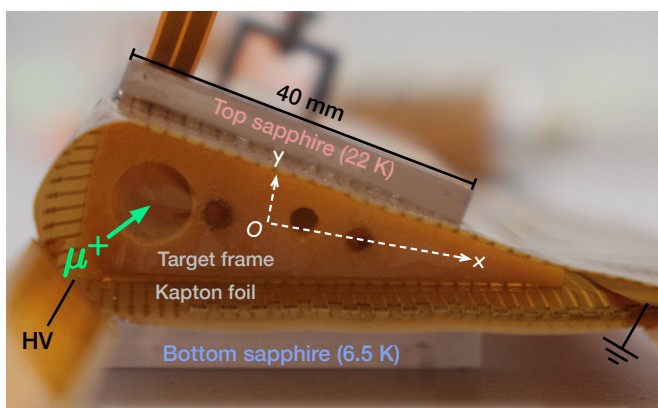

Figure 3: (Left) Picture of the Kapton foil, which, when folded, forms the top, bottom, and lateral walls of the target. The maximum positive voltage (HV) applied to the central large electrode is distributed to the other electrodes through two voltage dividers composed of SMD resistors. (Right) Picture of the helium gas target. The Kapton foil is folded and glued around the target frame to seal the target.

copper electrodes which are specially shaped and set to various high voltages to define the electric field with longitudinal and transverse components as shown in Fig. 2.

The highest positive voltage HV, ranging from 3 kV to 5 kV depending on the target pressure, is applied to the bottom left corner of the target (see Fig. 3 (Right)). The electric potential of the other electrodes is gradually reduced toward the target tip making use of voltage dividers, resulting in a transverse field at 45° with respect to the $x$-axis. The V-shape of the electrodes defined by the angle $\alpha$ in the region of $-30\,\text{mm} < z < 30\,\text{mm}$, as indicated in Fig. 3 (Left), serves to produce the longitudinal component of the electric field, that points toward the $z = 0$ plane. This angle $\alpha$ determines the relative strength between the longitudinal and the transverse E-fields, here $E_\perp/E_L = 2.8$.

Two sapphire plates are glued to the bottom and top target walls to ensure a uniform temperature across these surfaces. The bottom sapphire plate is thermally connected to a cold finger, which in turn is thermally linked to a pulse tube cooler (Cryomec PT-415 [33]) located outside the solenoid. The bottom sapphire plate also acts as an electrical insulator, isolating the target from the grounded cold finger. The top sapphire plate, on the other hand, is heated to 22 K using a resistive heat pad, creating a temperature (and density) gradient in the helium gas between the top and bottom sapphire plates.

We tested the mixed compression using a 13.6 MeV/c momentum beam delivered by the $\pi$E1 beamline at the CHRISP facility at PSI [34]. The momentum was chosen to maximize the rate of muons stopping in the 8 cm long helium gas target while avoiding excessive background in the detectors used to expose the compression. The muon beam was focused into the target by the combined effects of the last quadrupole triplet of the beamline and the fringe field of the solenoid producing the 5 T field. Each muon entering the setup at a stochastic time (cw beam) was detected by an entrance counter made from a 30 µm thick plastic scintillator read out by silicon photomultipliers (SiPM) and positioned close to the solenoid entrance face at the location of the first focus. The muon then propagated for about 40 cm before entering the target through a 25 µm thick Kapton window, passing also the cryostat vacuum window (25 µm), the thermoshield window (2.4 µm), and a 2 cm thick copper collimator through an aperture of 8 mm diameter.

The motion of the muons drifting in the helium gas target was monitored by positioning several plastic scintillators [28] around the target to detect positrons from muon decays. The

likelihood of these scintillators detecting decay positrons was strongly dependent on the position of the muon decay relative to each scintillator. Consequently, the motion of the muons within the target can be deduced by analyzing the time evolution of the count rates in these scintillators, with time $t = 0$ defined by the entrance counter.

Figure 4 (Top) shows the placement of the various scintillators around the target as side (Left) and top (Right) views. These scintillators (positron detectors) can be classified into two main categories: the transverse detectors (T1 and T2) used to expose the muons' motion along the $x$-axis, and the L1-L3 detectors that are sensitive to the muons reaching the target tip exposing at the same time longitudinal ($z$) and transverse ($y$) compressions. For reference the T1 and T2 detectors have a size of $(4 \times 4 \times 30)$ mm$^3$ while the bottom L1-L3 detectors are $(4 \times 4 \times 4)$ mm$^3$.

Figure 4 (Middle) and (Bottom) depict the detection efficiencies of these detectors, simulated with GEANT4, projected onto the $xy$ and $xz$-plane, respectively. The position-dependence of the detection probability, which results in a relatively good position resolution, was obtained by embedding the scintillators into a massive copper collimator and by organizing the scintillators in telescope-pairs read out in coincidence (top and bottom detectors). Each detector effectively covers distinct zones of the target volume. For instance, the L1-L3 detectors are only sensitive to muon-decays occurring at the tip of the target at their respective $z$-position. For reference Fig. 4 also indicates the position of the muons at time $t \approx 0$, i.e., the muon stopping distribution before compression takes place. To further increase the position resolution an energy threshold is applied to each positron counter, efficiently removing events from secondary particles. Each scintillator was optically coupled to a wavelength-shifting fiber to transport their optical signals from the cryogenic section of the setup to outside the cryostat. The fibers were then read out by SiPMs [28].

## 5 Measured time spectra and comparison to simulation

Revealing the drift of the muon beam within the target and demonstrating its compression are accomplished through the analysis of the time spectra observed in the various positron counters. In these time spectra, the reference time $t = 0$ corresponds to the moment when the muons pass the entrance counter. The counts in the time spectra are normalized to the number of muons passing the entrance counter, and are lifetime-compensated, meaning they are multiplied by $e^{+t/\tau_\mu}$ where $\tau_\mu = 2.2\,\mu$s is the muon lifetime.

The time spectra shown in Fig. 5 clearly prove that starting from the muon stopping distribution centered at $x \approx -15$ mm (see Fig. 4) at $t \approx 0$, the muons gradually drift in the $x$-direction over time. At $t \approx 0$, the muons are located in the muon stopping region, which falls outside the detection acceptance of the various detectors, resulting in zero or nearly zero counts in all three time spectra. Around $t \approx 2\,\mu$s, the muons pass through the region with the maximum acceptance for the T1 coincidence, as evidenced by the peak in the number of counts in the T1 coincidence time spectrum. After $t \approx 4\,\mu$s, they traverse the region of maximum acceptance for the T2 coincidence, leading to a peak in the T2 time spectrum. Finally, after approximately $5\,\mu$s, the muons reach the tip of the target and enter the acceptance region of the L2 coincidence resulting in a pronounced increase of counts in the L2 time spectrum. The decrease in counts for $t > 2.5\,\mu$s in the T1 time spectrum is due to muons leaving the region of maximal detection efficiency for the T1 coincidence, which is located around $x \approx -5$ mm for $y \approx 0$ (see Fig. 4). A similar decrease is observed in the T2 time spectrum for $t > 4\,\mu$s, indicating that muons pass through and eventually leave the region of maximal detection efficiency for the T2 coincidence, located around $x \approx 10$ mm. In contrast, the lifetime-compensated counts in the L2 spectrum remain constant at late times. Indeed, as shown in Fig. 4 (Right),

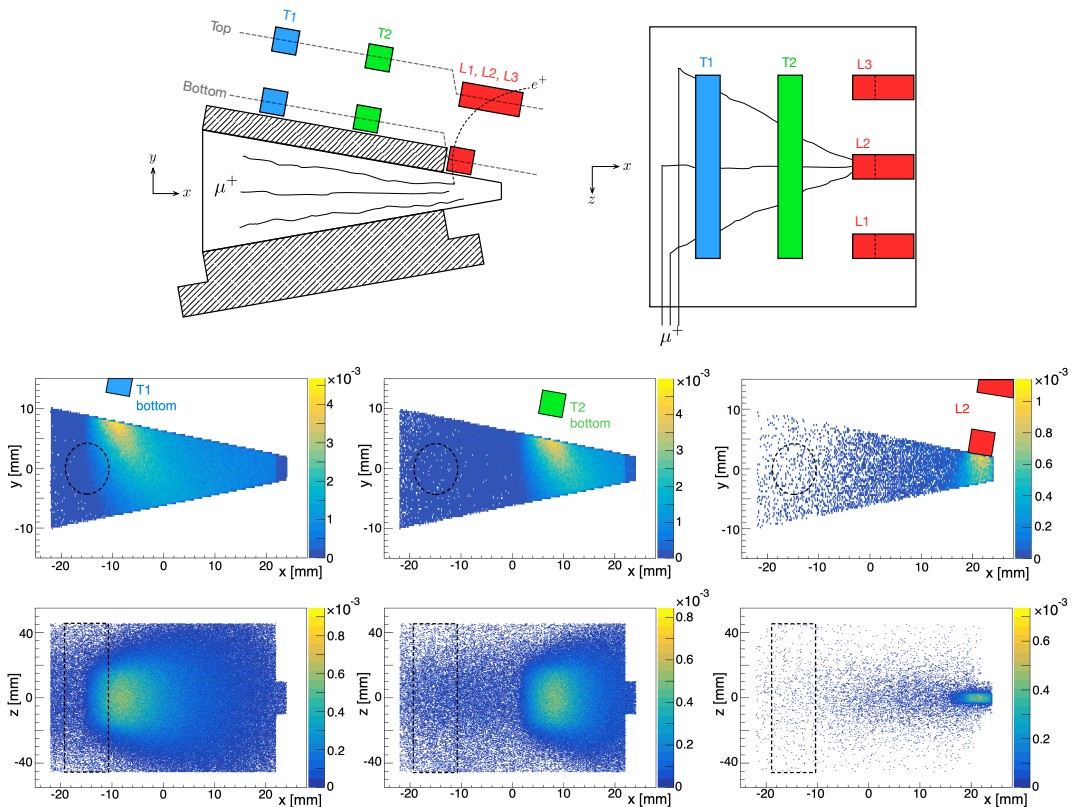

Figure 4: (Top) Placement of the scintillators (positron detectors) around the target projected onto the $xy$-plane (Left) and the $xz$-plane (Right). Each bottom detector is paired with a top detector to form a telescope. (Middle) Simulated position-dependent detection efficiencies projected onto the $xy$-plane for the various positron detectors. These detection probabilities were obtained by requiring coincidences between top-bottom detector pairs and including the shielding from a massive copper collimator between the detector pairs (not shown in the picture). The dashed black circles indicate the initial muon stop distribution. (Bottom) Similar to (Middle) but projected on the $xz$-plane. The dashed black rectangles indicate the initial muon stop distribution.

muons drifting in the $+x$-direction are unable to exit the region of highest detection efficiency (for the L2 coincidence) located at $x \approx 22$ mm, as the target ends up at $x \approx 24$ mm.

At late times, all muons must eventually come to rest on one of the target walls, such that the lifetime-compensated time spectrum becomes flat. Without transverse (i.e., $y$-direction) compression, most muons would hit the top or bottom target walls before reaching the tip of the target, resulting in much less L2 coincidence counts and a less pronounced decrease in the T1 and T2 signals. The large number of counts observed in the L2 spectrum therefore demonstrates not only that muons are drifting in the $x$-direction, but also that an efficient transverse compression is indeed taking place.

The onset of an effective longitudinal compression is well visible by comparing the L1-L3 time spectra of Fig. 6. Only the spectrum of the L2 coincidence, which is sensitive to muons around $z = 0$, shows an increase of counts at late times. The other two pairs, L1 and L3, which are placed at $z = \pm 18$ mm show basically flat time spectra at small count number. This demonstrates that, as muons drift in the $x$-direction, they are efficiently attracted to the central

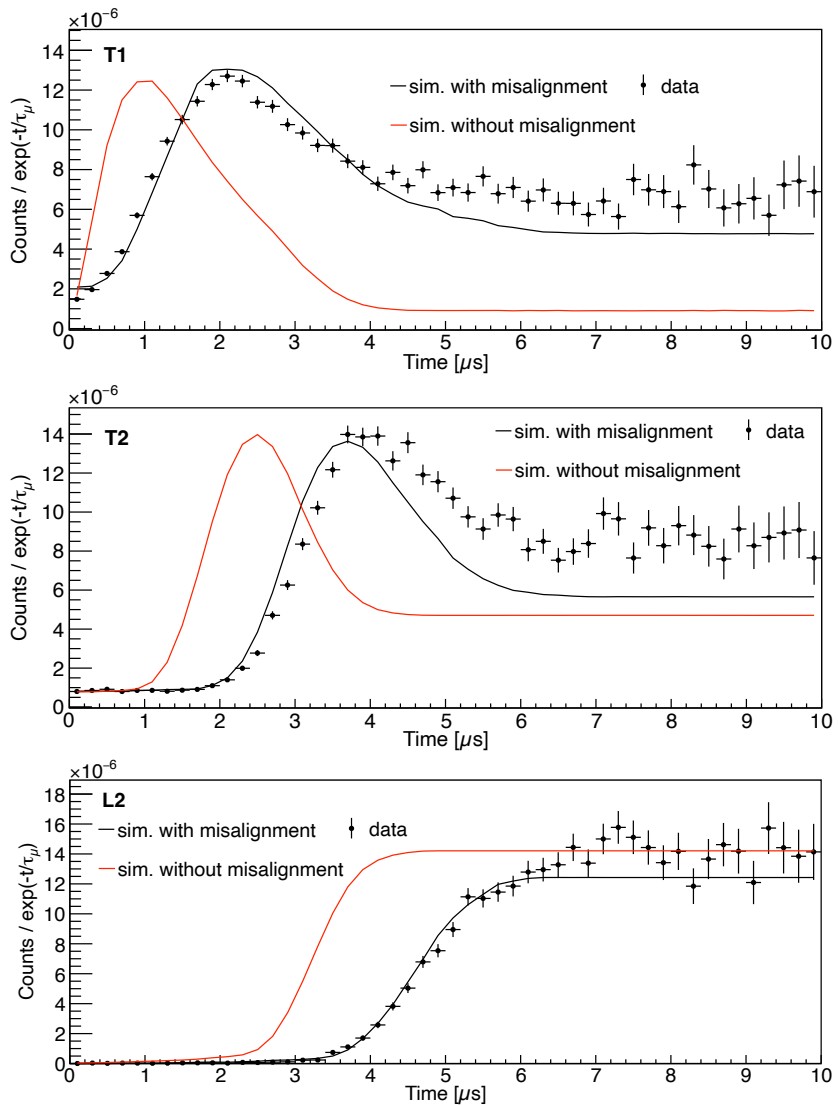

Figure 5: Measured time spectra in T1, T2 and L2 coincidences for a target with 10 mbar pressure, $T_{\text{top}} = 23.1\,\text{K}$, $T_{\text{bottom}} = 6.6\,\text{K}$, HV = +4.99 kV and 5 T B-field. The red curves represent the simulation at design conditions. They have been normalized to match the data as described in the main text. The black curves are simulations that assume a misalignment between target and B-field (beam) axes as in Fig. 7 (Right). Each black curve is fitted to the data using two free parameters: a normalization factor (amplitude) and an additional flat (muon-correlated) background.

plane of the target at $z = 0$, and do not reach or cross into the acceptance regions of the L1 and L3 detectors. Together with the substantial increase in counts at late times compared to $t \approx 0$ in the L2, this provides compelling evidence that efficient longitudinal compression is occurring in the $z$-direction as expected from simulations.

Summarizing, the behavior observed in the various time spectra qualitatively demonstrates efficient simultaneous compression in both transverse and longitudinal directions. To obtain more quantitative results, the observed time spectra have to be compared to simulations carried out using GEANT4 [24–26].

Simulations have been performed, accounting for the injection and stopping of the muon beam in the helium gas target, the muons' drift and compression within the target, and the sub-

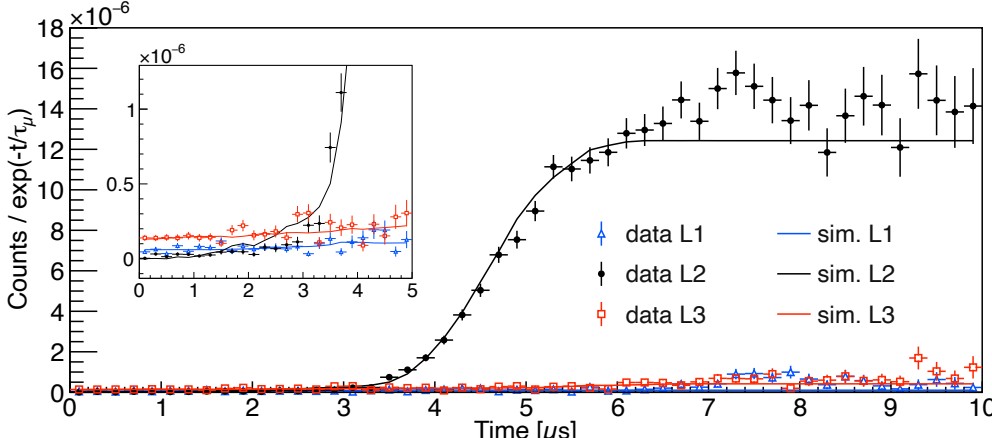

Figure 6: Measured time spectra as in Fig. 5 in the L1, L2 and L3 coincidences under the same target conditions. The curves represent simulations that assume a misalignment between target and B-field (beam) axes. The inset provides a zoomed-in view at early times.

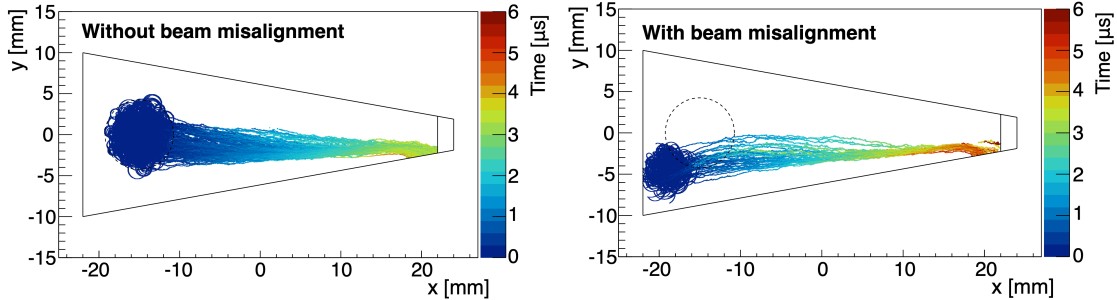

Figure 7: Simulated muon trajectories (projected to the $xy$-plane) for the target conditions of Fig. 5 for two muon stop distributions (at times $t \approx 0$). The black dashed circle indicates the projection of the target entrance window. (Left) The in-coupled muon beam is aligned with the target axis. Hence, the projected stopping distribution is aligned with the target entrance window. (Right) The stopping distribution of the muon is shifted toward negative $x$- and $y$-directions due to a misalignment between beam direction and target axis.

sequent detection of the decay-positrons. For muon kinetic energies above 1 keV and for the detection of the positrons, standard GEANT4 packages have been used (G4MuMultipleScattering, G4MuIonisation, G4eMultipleScattering, G4eIonisation, G4eBremsstrahlung and G4eplusAnnihilation [35]), while for the low-energy regime in the target we implemented customized low-energy processes as detailed above and in Ref. [26].

In Fig. 7 (Left), a simulation of muon trajectories in the target is presented under the conditions employed for the measurements of Fig. 5, where the largest statistics were accumulated. To generate the time spectra, the simulated trajectories (time-dependent positions of the muons) need to be combined with the position-dependent detection efficiencies of the various positron counters of Fig. 4. The resulting time spectra, shown in red in Fig. 5, deviate significantly from the measurements. Due to this substantial mismatch, we do not fit the simulations to the data. Instead, we simply overlay the simulations on the measurements, allowing for a normalization such that the maxima of the simulations align with the maxima of the measured time spectra. Additionally, we introduce a flat background (muon-correlated

background) so that simulations and measurements match at time $t = 0$.

It is important to note that the rescaling of the amplitude is primarily justified due to significant uncertainty in the muon stopping probability within the helium gas target that strongly depends on the momentum distribution of the muons entering the target which is not well known. Moreover, this rescaling of the amplitude absorbs the uncertainties related to the absolute efficiencies of the positron detectors, which includes energy threshold effects. The introduction of an additional flat background could be attributed to factors such as more muons stopping at the target walls than anticipated and uncertainties in the small energy deposition for muon decays occurring far from the scintillators.

While qualitatively the measured time spectra show an efficient mixed compression of the muon beam, a detailed comparison with simulations reveals that muons reach the target tip with an additional delay of approximately 2 µs relative to simulations. Since it was not feasible to experimentally investigate the origin of this discrepancy within the limited time available during data taking at the CHRISP facility, we investigated the potential source of this deviation through GEANT4 simulations.

We explored several factors that could account for the discrepancy, including variations in the $\mu^+$-He total elastic cross-sections, changes in the helium density due to fluctuations in the pressure or temperature distribution, and a potential decrease of the electric field strength caused by accumulation of static charges on the insulating part of the target. We also examined variations in the relative positioning between the detectors and the target, as well as different energy thresholds applied for the positron counters. Additionally, we considered the impact on the time spectra of possible hydrogen contamination in the helium gas, which could increase the neutralization of $\mu^+$ into muonium. However, none of these variants allowed to overcome the observed discrepancies between simulated and measured time spectra.

Finally, we investigated the effect of a displaced muon stop distribution (muon position at time $t \approx 0$) relative to the design value. Although the target frame (entrance window) and a collimator placed 10 cm upstream limit the maximal possible displacement, significant misalignment, both tilt and shift of the target axis relative to the magnetic field axis, was still possible.

Simulations show that such misalignments can considerably influence the observed time spectra. Specifically, the black time spectra in Fig. 5, simulated assuming the initial stop distribution shown in Fig. 7 (Right), display a markedly improved agreement with the measurements. Indeed, by misaligning the stop distribution of the muon beam as shown in Fig. 7 (Right) toward smaller $x$- and $y$-values causes the muons to travel a longer path while moving through a region of higher gas density compared to the properly aligned case. As a result, the time spectra shown in black in Fig. 5 exhibit a shift to later times compared to the red curves of the aligned case. These time spectra were fitted to the measurements using two free parameters for each detector pair (T1, T2, and L2): the first is a normalization to account for uncertainties in the absolute efficiency of the detector-pairs and the muon stopping probability, the second is a flat (muon-correlated) background to address the effects of muons stopping at the target endcaps and the uncertainty in energy deposition within the small scintillators embedded in the large copper collimators.

Similarly, Fig. 8 presents measured and simulated time spectra for the same three-detector pairs —T1, T2, and L2 — but for a 5 mbar target pressure (compared to the 10 mbar of Fig. 5) and for variations of the HV settings. The same misalignment used in Fig. 5 was assumed in the simulations. In this case the two fits in each panel share a common normalization and a common additional muon-uncorrelated background.

Figure 9 shows measured and simulated time spectra for variations of the magnetic field strengths (5 T and 4 T) at a target pressure of 8 mbar. The data confirms that the compression is becoming faster with decreasing B-field strengths as observed in the simulations. Also in

this case, the same fitting procedure and misalignment, as used in Fig. 5, were applied.

While the presented fits are qualitative, it is important to emphasize that introducing this misalignment between target and magnetic field axes—while keeping all other parameters at their design values—has led to a satisfactory agreement between simulations and measurements. This agreement was achieved across three different detector pairs under various target conditions, including three different pressures, two magnetic field strengths, and several HV settings.

Given that we obtain a satisfactory agreement at multiple conditions and multiple detectors simply by introducing this misalignment, we decided against further engaging in a multidimensional (beam-target misalignment, effective E-field, distortion of the target geometry, $\mu^+$-He elastic cross section, He pressure, scintillator position, etc.) optimization. This decision is based on the understanding that such optimization is extremely time-consuming, the results would be not unique and would likely yield limited additional insights. Furthermore, our setup, by its design, is susceptible to relatively large misalignments between the target, the solenoid and the beamline making our solution plausible. The stop distribution shown in Fig. 7 (Right) that explains well all measurements can be obtained by a shift of the muon beam by 3 mm in both negative $x$- and negative $y$-directions in conjunction with a tilt of 2.3° between magnetic field and target axes. This angle is well within the maximal acceptance of about 4° of the collimator-target-window system and remains within the possible range of solenoid-target tilts.

A precise determination of the muon beam compression efficiency, defined as the number of muons stopping in the target and passing through a $\Delta y \times \Delta z = 1 \times 1.3$ mm$^2$ spot at the target tip (corresponding to the planned size of the orifice), from data or by comparing measured and simulated time spectra, is not feasible. This is primarily due to the inherent challenges to precisely determine the position of the muon upon arrival at the target tip and the substantial uncertainty associated with the muon stopping probability in the helium gas target.

Yet, from these measurements, it is possible to roughly estimate the fraction of stopped muons that reach the region of maximal acceptance of the L2 coincidence. This estimation can be made by comparing the ratio of counts at late times to counts at early times in the L2 time spectra, as obtained from both simulations and measurements. The advantage of this method is that it is largely independent of the muon stopping probability and certain aspects related to detection efficiency. The ratio obtained from simulations closely matches the ratio from the measured time spectra, indicating that the simulations and measurements are consistent.

Leveraging the simulation (rectifying for the beam-target misalignment, slightly extending the target and controlling the temperature at the target tip), we have calculated that muon beam compression for an initial stopping volume in the shape of a cylinder with a diameter of 1 cm and a length of 5 cm to a size of $\Delta y \times \Delta z = 1 \times 1.3$ mm$^2$ can be achieved within 5 μs with an efficiency of about 90% (excluding muon decay).

## 6 Conclusion

A significant milestone in the muCool beam development was reached by demonstrating efficient mixed compression for the first time. The observed time spectra provide clear evidence of muons efficiently drifting within the cryogenic helium gas target while undergoing simultaneous compression both in transverse ($y$) and longitudinal ($z$) directions.

The comparison between measured and simulated time spectra using nominal values for the target parameters revealed a significant discrepancy. However, this discrepancy can be largely resolved by fine-tuning the muon beam direction with respect to the target axis.

Although the detection system did neither allow for the precise measurement of the com-

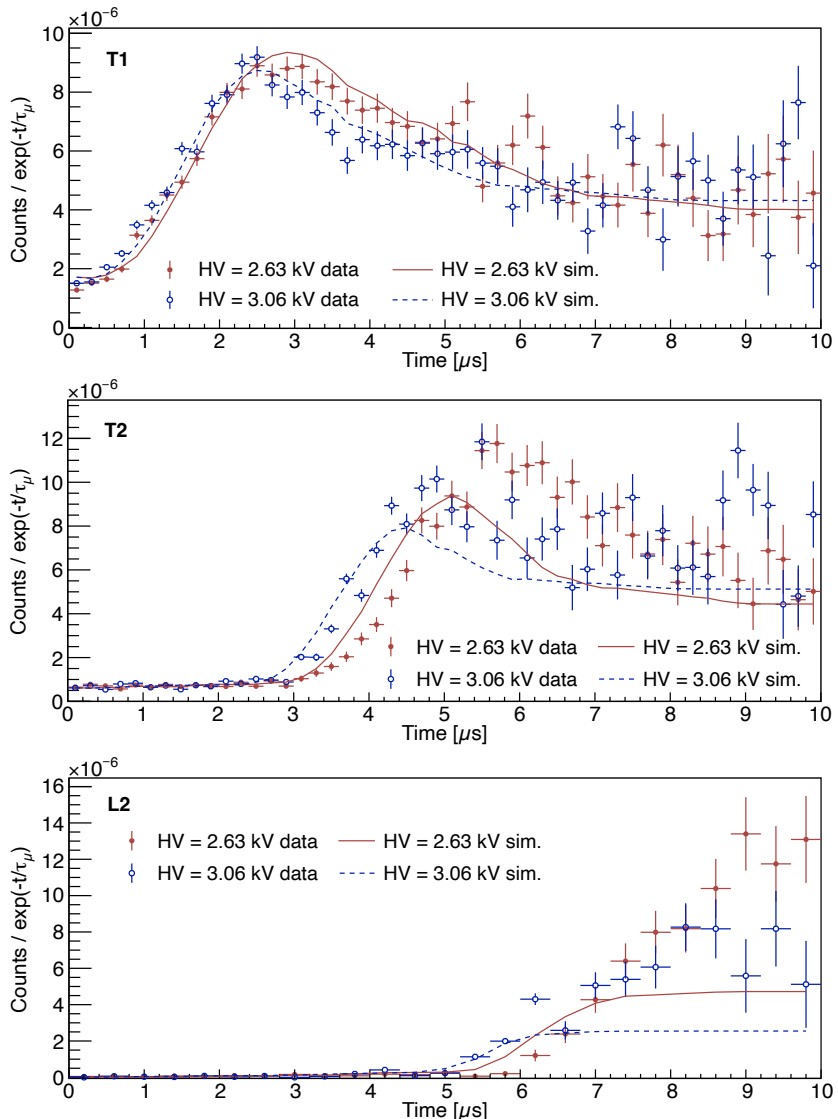

Figure 8: Measured time spectra in the various detector-pairs for a target pressure of 5 mbar, a 5 T field strength and two HV configurations. Both the red and blue curves are simulated time spectra assuming a misalignment between target and magnetic field axes as in Fig. 7 (Right). The two fits in each panel share a common normalization and a common additional muon-uncorrelated background.

pression efficiency nor the determination of the final spot size, the agreement ultimately achieved between simulation and experiment validates both our experimental technique and the simulations, predicting a 90% compression efficiency (excluding muon decay) within 5 μs. This efficiency allows us to estimate the overall performance of the muCool compression scheme. The total efficiency is estimated by considering each sequential stage — starting from muon injection into the solenoid and target, followed by compression, extraction, and final re-acceleration — with the following approximate efficiencies for each step as obtained from GEANT4 simulations: off-axis injection into the solenoid ($55 \pm 5\%$); transmission through the target entrance window ($50 \pm 5\%$); muon stopping probability in the active region of the helium gas target ($0.6 \pm 0.1\%$); compression efficiency, excluding muon decay ($90 \pm 10\%$); survival during the $\sim 5$ μs compression time, accounting for muon decay ($8 \pm 1\%$); extraction efficiency, excluding muon decay ($80 \pm 10\%$); survival during the $\sim 1.5$ μs time needed for extraction

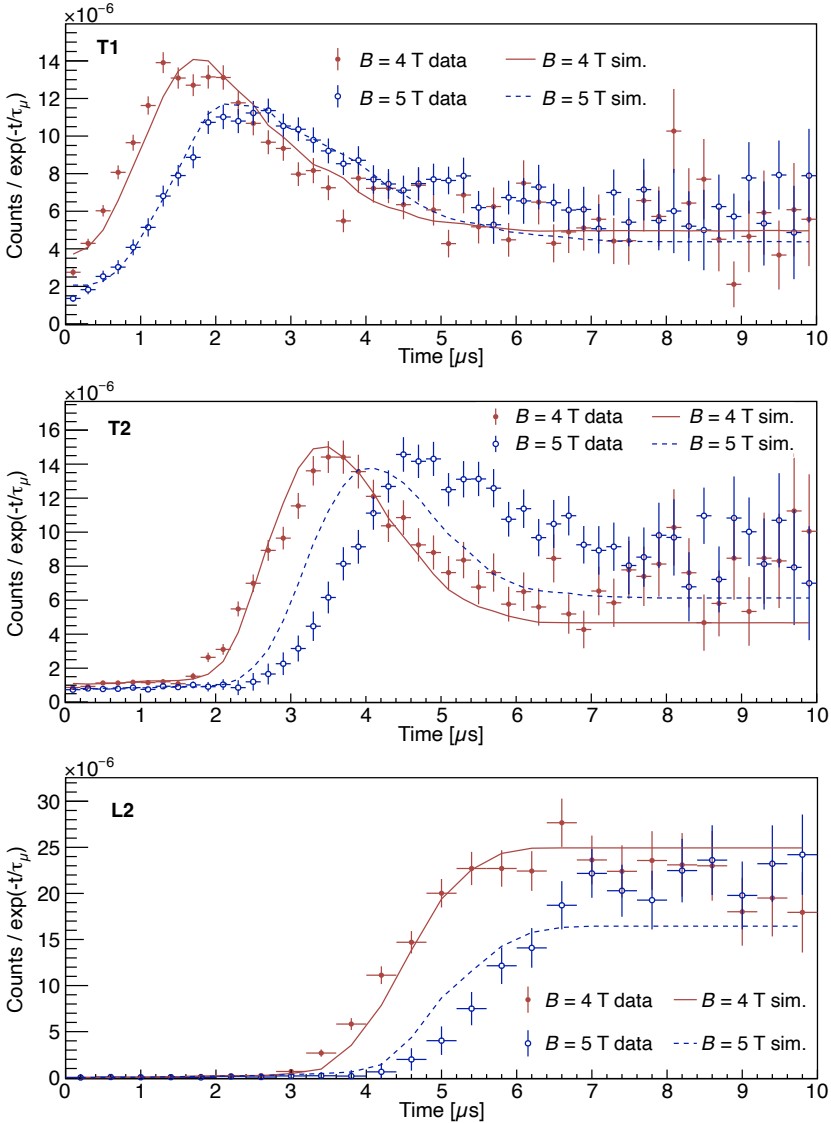

Figure 9: Measured time spectra in the various detector-pairs for a target pressure of 8 mbar and HV = 4.16 kV and two values of the magnetic field strength. Each simulated spectrum is fit to data with two free parameters, an amplitude and an additional flat (muon-correlated) background, and assumes a misalignment as in Fig. 7 (Right).

and propagation to the re-acceleration stage ($50 \pm 10\%$); coupling into the re-acceleration stage ($80 \pm 10\%$); and re-acceleration to 10 keV followed by extraction from the magnetic field ($50 \pm 10\%$). Multiplying these factors yields an overall efficiency of $1.9(8) \times 10^{-5}$. This estimate assumes operation with the HIMB beam [20, 21] at a momentum of 28 MeV/$c$ and a target performance as obtained in this study, operated at 7 mbar. One of the dominant uncertainties arises from the extraction from the magnetic field, conservatively estimated at $(50 \pm 10)\%$. This process has so far only been studied at a very preliminary level and depends critically on the beam quality at the ferromagnetic plate. A more reliable estimate will require, among other steps, the implementation of dedicated collisional processes in the re-acceleration region in the keV regime, as well as detailed studies of the lenses that must refocus the beam after the position-dependent azimuthal re-acceleration at the ferromagnetic plate [36]. The overall efficiency can be improved by enhancing the target performance, for example by en-

larging the active region, optimizing the electric field, refining the gas density distribution, and employing an array of muCool targets. Importantly, the overall efficiency is highly input-beam-dependent: replacing the HIMB beam with the $\pi$E1 beam boosts it by a factor of five. This boost arises from the smaller momentum bite (enhancing stopping probability), reduced transverse momentum (minimizing reflections in the solenoid), and smaller beam size (improving transmission through the entrance window) of the $\pi$E1 beam. Efficiencies up to $10^{-4}$ therefore appear within reach for the muCool scheme.

The upcoming phase of the muCool project involves extracting the compressed muons from the cryogenic helium gas target into a vacuum. This critical step will enable a more precise quantification of the overall efficiency of the muCool scheme. For this, we will also modify the beam alignment procedure and mechanical design to solve the misalignment issue encountered in these measurements.

Preliminary simulations of the complete scheme — including beam extraction from the target, re-acceleration, and transport along the magnetic field lines — yield the output beam parameters at the magnetic shield position (with a 0.1 T field), as summarized in Fig. 1. When compared to the High Intensity Muon Beam (HIMB) parameters [20,21] these results indicate up to a factor of $10^9$ improvement in the six-dimensional phase space.

# Acknowledgments

We acknowledge technical support by Alexey Stoykov, Adamo Gendotti, Florian Barchetti, Davide Reggiani, Urs Greuter, and the PSI operation groups for providing excellent running conditions at the $\pi$E1 beamline.

**Funding information** We acknowledge support from the Swiss National Science Foundation (SNSF) grant numbers 200441 and 172639.

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
