# Peer review of "Phase space compression of a positive muon beam in two spatial dimensions"

_SciPost Physics Core, doi:SciPost Phys. Core 8, 071 (2025)_

## Round 1 · Referee Report · Anonymous (Referee 1) · 2025-7-9

Strengths

This paper focuses on an experimental demonstration of low energy positive muon beam phase space compression in two dimensions using a cryogenic helium gas target with a density gradient, a homogeneous magnetic field and an electric field. Such an arrangement could benefit muSR, and atomic and particle physics measurements which require small transverse phase space, low energy, and high intensity including proposed gravity and laser spectroscopy experiments. A stated
Goal is to reduce the phase space of a typical “surface” positive muon beam
by a factor of 109 with an efficiency of up to 10−4with tunable energy in the keV regime, with anenergy spread of less than 100 eV and a sub-millimeter transverse size.
Qualitative experimental validation of muon beam compression in the helium
gas target is demonstrated with a muon beam of13.6 MeV/c (instead of 28 MeV/c) and GEANT4 simulations show rough correspondence with data when various assumptions are made concerning possible beam and magnetic field misalignments.

The paper reasonably describes the motivation, theoretical basis, experimental design, and results, and presents conclusions confirming the observation of some 2-D phase space compression.

Weaknesses

Some aspects of the discussion could be clarified. See comments below.

Report

The paper is suitable for publication in this journal after the authors consider the comments below.

Requested changes

Comments
The components of the rather crude position measuring apparatus are somewhat awkwardly named (e.g. Trans 2, Tile 1). Renaming them with some functional or position nomenclature might make the discussion of results more readable.

Continuation of the tile timing simulation in fig. 5 beyond 6 μs would be interesting to see; alternatively, provide an explanation for the cut-off.
It would be useful to further explain the reference to muons "flying by". Is something missing from the simulation causing the disagreement at later times; even with the introduction of extra parameters the agreement of the simulations with data isn’t great.

It appears that fig. 6 contains the same data for Tiles 2,5 as fig. 5; were simulation done for the other tiles indicating some level of agreement? In Fig. 9 the B=5 T data/simulations disagree by a lot; is there an explanation?

The conclusions present an estimate of the overall compression efficiency; it would be useful to show the factors and uncertainties that went into those estimates.

Recommendation

Publish (easily meets expectations and criteria for this Journal; among top 50%)

  • validity: high
  • significance: good
  • originality: high
  • clarity: good
  • formatting: excellent
  • grammar: excellent

Author:  Ryoto Iwai  on 2025-08-19  [id 5741]

(in reply to Report 1 on 2025-07-09)

Dear Editor, Dear Reviewer,

We sincerely appreciate your review of our paper. The attached file contains our responses to your comments

Best regards
muCool collaboration

Attachment:

Reply_to_editor.pdf

---

## Round 3 · Referee Report · Anonymous (Referee 1) · 2025-9-12

Strengths

The authors have responded to the previous comments by providing estimated uncertainties to the performance projections.

Report

I recommend publication.

Recommendation

Publish (meets expectations and criteria for this Journal)

---

## Editorial Decision

published